# Exploring the Therapeutic Potential of Mitragynine and Corynoxeine: Kratom-Derived Indole and Oxindole Alkaloids for Pain Management

**DOI:** 10.3390/ph18020222

**Published:** 2025-02-06

**Authors:** Ahmed S. Alford, Hope L. Moreno, Menny M. Benjamin, Cody F. Dickinson, Mark T. Hamann

**Affiliations:** Department of Drug Discovery and Biomedical Sciences, College of Pharmacy, Medical University of South Carolina, 70 President St., Charleston, SC 29425, USAhamannm@musc.edu (M.T.H.)

**Keywords:** mitragynine, corynoxeine, pain management, *Mitragyna speciosa*, systematic review, pharmacology, β-arrestin, opioids

## Abstract

The search for effective pain management solutions remains a critical challenge, especially amidst growing concerns over the use of conventional opioids. In the US, opioid-related mortality rates have surged to as many as 80 deaths per 100,000 people in some states, with an estimated economic burden of USD 1.5 trillion annually—exceeding the gross domestic product (GDP) of most US industrial sectors. A remarkable breakthrough lies in the discovery that indole and oxindole alkaloids, produced by several genera within the plant Tribe Naucleeae, act on opioid receptors without activating the beta-arrestin-2 pathway, the primary driver of respiratory depression and overdose deaths. This systematic review explores the pharmacological properties, mechanisms of action, dosing considerations, interactions, and long-term effects of mitragynine and corynoxeine, alkaloids from the Southeast Asian plant Mitragyna speciosa (kratom) and others in the Tribe Naucleeae. Mitragynine, a partial opioid receptor agonist, and corynoxeine, known for its anti-inflammatory and neuroprotective effects, demonstrate significant therapeutic potential for managing diverse pain types—including neuropathic, inflammatory, nociceptive, visceral, and central pain syndromes—with a focus on cancer pain. Unlike traditional opioids, these compounds do not recruit beta-arrestin-2, avoiding key adverse effects such as respiratory depression, severe constipation, and rapid tolerance development. Their distinct pharmacological profiles make them innovative candidates for safer, non-lethal pain relief. However, challenges persist, including the unregulated nature of kratom products, inconsistencies in potency due to crude extract variability, potential for misuse, and adverse drug interactions. Addressing these issues requires establishing standardized quality control protocols, such as Good Manufacturing Practices (GMP), to ensure consistent potency and purity. Clear labeling requirements with dosage guidelines and warnings should be mandated to ensure safe use and prevent misuse. Furthermore, the implementation of regulatory oversight to monitor product quality and enforce compliance is essential. This review emphasizes the urgency of focused research to optimize dosing regimens, characterize the pharmacodynamic profiles of these alkaloids, and evaluate long-term safety. By addressing these gaps, the mitragynine- and corynoxeine-related drug classes can transition from promising plant-derived molecules to validated pharmacotherapeutic agents, potentially revolutionizing the field of pain management.

## 1. Introduction

The search for effective pain management solutions remains a critical challenge, especially amidst growing concerns over the side effects and dependency risks associated with conventional opioids. In the United States, opioid misuse has led to significant mortality and economic burden, with opioid-related deaths estimated at 80 per 100,000 individuals in some states and an annual cost of approximately USD 1.5 trillion [1]. Amid this crisis, certain indole and oxindole alkaloids from the plant tribe Naucleeae have emerged as promising alternatives due to their ability to engage opioid receptors without activating the β-arrestin-2 pathway, a key mediator of respiratory depression and other severe opioid side effects [2].

The alkaloid composition of *Mitragyna speciosa* (kratom) is dominated by mitragynine, which constitutes approximately 66.2% of the total alkaloid content in commercial kratom products. In contrast, 7-hydroxymitragynine, despite its higher potency at opioid receptors, is a minor component, accounting for only 0.01–0.03% of the total alkaloids. Other significant alkaloids include speciociliatine (8.6–16.6%), paynantheine (9.0–16.0%), and speciogynine (6.6–8.6%), with trace amounts of additional alkaloids such as corynoxeine, isocorynoxeine, and speciophylline. These secondary alkaloids may also contribute to kratom’s pharmacological profile but in a less pronounced manner. The variability in alkaloid composition across kratom products is influenced by factors such as genetic differences, environmental conditions, and processing methods, resulting in inconsistencies that affect their pharmacological effects and safety profiles.

Among these compounds, mitragynine and corynoxeine, derived from the Southeast Asian plant *Mitragyna speciosa* (Figure 1), commonly known as kratom, stand out as potentially safer alternatives to traditional opioids. While kratom contains a range of bioactive alkaloids, the oxindoles, including corynoxeine and its stereoisomer isocorynoxeine, are minor yet significant tetracyclic oxindole alkaloids. Kratom also contains other active compounds, such as flavonoids, polyphenols, and terpenoids, which contribute to the pharmacological effects of the raw plant products and crude plant extract. These oxindoles also occur in higher concentrations in other plants, such as species within the *Uncaria* genus, further highlighting their pharmacological importance [3]. Mitragynine, an indole alkaloid, is recognized as a partial agonist of the mu-opioid receptor (MOR), whereas corynoxeine demonstrates notable anti-inflammatory and neuroprotective properties. Unlike conventional opioids, both mitragynine and corynoxeine lack β-arrestin recruitment, which reduces risks of adverse effects such as respiratory depression, constipation, and tolerance development [4]. This unique pharmacological profile positions them as promising candidates for managing various types of pain—including neuropathic, inflammatory, nociceptive, visceral, and central pain syndromes—with a particular focus on cancer-related pain.

This review focuses specifically on mitragynine and corynoxeine as representative compounds of the indole and oxindole alkaloid classes, respectively. Although kratom is traditionally consumed in its whole form (e.g., crude leaves or powdered leaves), isolating and examining these two compounds allows for a targeted analysis of their unique therapeutic potential, mechanisms of action, and safety profiles. By focusing on mitragynine and corynoxeine, this review seeks to reflect the broader pharmacological potential of indole and oxindole alkaloids, addressing their potential applications in pain management without the confounding factors present in whole-plant kratom formulations. For reference, Table 1 and Table 2 present the principal structures of the oxindole and indole alkaloids derived from *Mitragyna speciosa*, providing an overview of these compounds’ structural diversity relevant to their pharmacological profiles.

Despite these promising pharmacological profiles, challenges remain, particularly around the unregulated nature of kratom products, variability in dosage, and potential for abuse. This review aims to identify and critically assess relevant studies to provide an in-depth understanding of the current evidence surrounding mitragynine, corynoxeine, and other kratom-derived alkaloids. While no clinical trials have yet defined therapeutic doses or established guidelines, the preclinical and observational data reviewed here may offer preliminary insights that could inform future research. This synthesis is intended to support clinicians and researchers in gaining a foundational understanding of these compounds’ potential pharmacological profiles and mechanisms, and to encourage further investigation into their role in pain management and other therapeutic areas. Policymakers and the general public may also find this review beneficial in understanding the complex landscape of kratom’s benefits and risks as they relate to public health considerations.

## 2. Results

### 2.1. Pharmacodynamics

Corynoxeine’s mechanisms of action suggest significant potential in the treatment of pain through several pathways:

Anti-Inflammatory Agent: Corynoxeine’s role as an anti-inflammatory agent is critical in managing pain linked to inflammation, a common cause in conditions such as arthritis and autoimmune diseases. Through inhibition of vascular smooth muscle cell (VSMC) proliferation, particularly by blocking the extracellular signal-regulated kinase (ERK1/2) phosphorylation pathway, corynoxeine reduces inflammatory processes in vascular and tissue contexts. This anti-inflammatory activity can help mitigate pain by lessening the release of pro-inflammatory cytokines and mediators that typically heighten pain sensitivity. By targeting inflammation pathways, corynoxeine supports tissue health and reduces the potential for chronic pain development [5,6].

Neuroprotective Actions and Autophagy Enhancement: Corynoxeine serves as a neuroprotective agent, offering significant potential in conditions associated with neuropathic pain where nerve cell damage or dysfunction contributes to pain perception. Through the enhancement of autophagy, corynoxeine helps clear neurotoxic protein aggregates such as alpha-synuclein, which can otherwise accumulate and cause neural stress and degeneration. By engaging pathways like Akt/mTOR to induce autophagy, corynoxeine may reduce neuronal death and maintain nerve function, effectively lowering pain signals associated with damaged or impaired nerves. This neuroprotective mechanism could be beneficial in conditions like peripheral neuropathy and neurodegenerative diseases, where nerve protection aligns with pain mitigation [6,7].

Calcium Channel Blockade and Vasorelaxation: Corynoxeine induces vasorelaxation by blocking L-type calcium channels in vascular smooth muscle cells, thus preventing the influx of calcium ions. Calcium ions play a critical role in smooth muscle contraction; by limiting their entry, corynoxeine reduces smooth muscle tone and relaxes blood vessels. This vasodilation effect is particularly beneficial in conditions where vascular constriction is linked to pain, such as migraines and certain vascular pain syndromes. Enhanced blood flow from reduced vascular resistance can alleviate symptoms by improving oxygenation and nutrient delivery to affected tissues, thus reducing ischemic or tension-induced pain [6,7,8].

Adrenergic Receptor Antagonism: By inhibiting α1A adrenergic receptors, corynoxeine further promotes vasodilation, countering the effects of adrenergic-mediated vasoconstriction. Adrenergic receptors, when activated, increase vascular tone, which can exacerbate pain in conditions like tension headaches and hypertension-related discomfort. Corynoxeine’s antagonism of these receptors allows for relaxation of blood vessels, which can ease pain by reducing the stress and pressure within the vascular system. This mechanism adds a layer of vascular relief in pain conditions where adrenergic activity contributes to vascular tightness and discomfort [9,10].

Potassium Channel Activation and Vascular Smooth Muscle Relaxation: As a potential potassium channel opener, corynoxeine hyperpolarizes vascular smooth muscle cells, further reducing cellular excitability and promoting vasodilation. The activation of potassium channels encourages the efflux of potassium ions, stabilizing the cellular membrane potential away from depolarization thresholds that trigger contraction. This action complements calcium channel inhibition, creating a dual mechanism for smooth muscle relaxation. By decreasing vascular resistance and enhancing blood flow, corynoxeine’s potassium channel activation may help alleviate pain from vascular sources, as seen in migraine and other vascular pain disorders [6,7].

Prevention of Vascular Smooth Muscle Cell Proliferation: Corynoxeine also inhibits the proliferation of vascular smooth muscle cells (VSMCs), an effect particularly relevant in conditions like atherosclerosis and restenosis, where abnormal cell growth contributes to vascular narrowing and pain. By blocking the ERK1/2 pathway, corynoxeine disrupts the signaling required for VSMC proliferation, which can prevent or slow down the progression of vascular blockages. This mechanism is essential in managing chronic pain linked to vascular diseases, as maintaining healthy, unobstructed vessels can reduce ischemic pain and improve blood circulation to affected regions [7,8].

In summary, corynoxeine’s multi-faceted mechanisms—including its roles as an anti-inflammatory agent, neuroprotective agent, calcium channel blocker, adrenergic receptor antagonist, and potassium channel opener—highlight its potential in effectively managing and treating various types of pain.

Mitragynine’s mechanisms of actions also suggest significant potential in the treatment of pain but through unique polypharmacology pathways (summarized in Figure 2).

Mu-Opioid Receptor (MOR) Partial Agonist: Mitragynine acts as a partial agonist at mu-opioid receptors (MOR), which are primary targets for traditional opioid analgesics. With a binding affinity of approximately 0.233 μM, mitragynine can activate MOR to produce analgesic effects but with limited efficacy compared to full agonists like morphine. This partial agonism allows mitragynine to provide pain relief while significantly reducing the side effects typically associated with opioid receptor activation, including respiratory depression and high potential for addiction. The decreased ability to fully activate MOR is linked to a reduced risk of tolerance development, meaning that patients are less likely to need increasing doses over time to achieve the same level of pain relief. Additionally, the partial agonism at MOR may result in lower risks of gastrointestinal side effects, such as constipation, which is a common issue with traditional opioids [9,10,11].

Kappa-Opioid Receptor (KOR) Antagonist: In contrast to its partial agonism at MOR, mitragynine functions as an antagonist at kappa-opioid receptors (KOR). KOR activation is often associated with dysphoria, hallucinations, and a reduction in reward-seeking behaviors. By antagonizing KOR, mitragynine reduces these adverse effects, potentially providing mood-stabilizing effects and lessening dysphoric reactions. KOR antagonism also contributes to pain relief, as KOR can play a role in modulating pain perception, especially in conditions where stress and mood are exacerbated by pain. This dual MOR agonism and KOR antagonism may offer a balanced analgesic effect while avoiding some of the psychological side effects of KOR agonists [12].

Competitive Antagonist at Serotonin Receptors (5-HT2A): Mitragynine acts as a competitive antagonist at the 5-HT2A serotonin receptor, which modulates the release of various neurotransmitters involved in mood, cognition, and pain perception. By blocking 5-HT2A receptors, mitragynine can reduce serotonin-related neurotransmission, which may help in stabilizing mood and controlling pain perception, particularly in individuals with anxiety or depression-related pain. This antagonism is also thought to reduce anxiety and prevent overstimulation of neural pathways, contributing to an improved quality of life for those with chronic pain [9,10].

Partial Agonist at 5-HT1A: As a partial agonist at the 5-HT1A receptor, mitragynine can enhance serotonin signaling to some extent, particularly in ways that are anxiolytic and mood-enhancing. The 5-HT1A receptor is linked to improved mood and anxiety reduction, both of which are beneficial in chronic pain management, as pain can be exacerbated by psychological stress and anxiety. By partially activating this receptor, mitragynine may improve the psychological state of individuals suffering from chronic pain, which indirectly aids in pain management by reducing pain perception linked to stress and anxiety [9,10].

Binding to 5-HT2B and 5-HT2C Receptors: Mitragynine has weaker interactions with 5-HT2B and 5-HT2C serotonin receptors, binding with a Ki of approximately 1260 nM. While these interactions are less pronounced than its actions on other receptors, they may have subtle modulatory effects on mood and cognition. Since 5-HT2B and 5-HT2C receptors are involved in the regulation of mood and appetite, the slight binding affinity could contribute to mitigating symptoms like irritability or loss of appetite in individuals with chronic pain, although the impact is likely minor compared to its effects on other receptors [9,10].

Alpha-2 Adrenergic Receptor Agonist: Acting as an agonist at alpha-2 adrenergic receptors, mitragynine engages mechanisms similar to clonidine, a drug used to manage opioid withdrawal symptoms. Alpha-2 agonism can reduce norepinephrine release, which calms the sympathetic nervous system and provides a sense of relaxation and reduced arousal. This agonism is particularly useful for managing symptoms of opioid withdrawal, including cravings, irritability, and heightened pain sensitivity. The activity at alpha-2 receptors may also contribute to the overall analgesic effect of mitragynine by reducing sympathetic responses that can intensify pain perception in stressful situations [9,13].

Dopamine D2 Receptor Affinity: Mitragynine’s affinity for dopamine D2 receptors suggests potential antipsychotic effects, which may alleviate symptoms of psychosis in vulnerable individuals. By interacting with D2 receptors, mitragynine could theoretically reduce excessive dopamine activity, which is linked to conditions such as schizophrenia and bipolar disorder. For chronic pain patients, this dopaminergic modulation might help in managing pain perception and emotional responses, as well as in mitigating the negative psychological symptoms sometimes associated with chronic pain, such as anhedonia and low motivation [14].

β-Arrestin Activity: Unlike traditional opioids, mitragynine does not recruit β-arrestin 2 at the mu-opioid receptor, which is significant because β-arrestin 2 recruitment is linked to several adverse effects of opioids, including respiratory depression, constipation, and tolerance [15]. The absence of β-arrestin 2 recruitment in mitragynine’s mechanism suggests a potentially safer side effect profile, offering effective pain relief without the heightened risks associated with conventional opioids. This property is a crucial factor in its unique pharmacological safety, making it a promising candidate for further research in pain management settings.

Cannabinoid Receptor Modulation: Mitragynine also interacts with cannabinoid receptors, particularly CB1 and CB2, which play roles in modulating pain and inflammation. Research indicates that mitragynine’s analgesic effects, particularly in neuropathic pain, may be mediated through these cannabinoid pathways. For instance, in models of chemotherapy-induced peripheral neuropathy, the analgesic effect of mitragynine was lessened when cannabinoid receptors were blocked, suggesting that these receptors are instrumental in its effect on neuropathic pain. This cannabinoid receptor interaction provides a unique avenue for pain relief, especially in complex pain conditions involving both central and peripheral mechanisms [16].

TRPV1 Modulation: Recent studies have shown that mitragynine may modulate the Transient Receptor Potential Vanilloid 1 receptor (1), a non-opioid pathway involved in sensing noxious stimuli, such as heat and inflammation. By affecting TRPV1 receptors, mitragynine can potentially reduce pain signaling in inflammatory and neuropathic pain conditions. TRPV1 modulation is particularly relevant for pain states where peripheral sensitization occurs, as blocking or modulating these receptors decreases the activation of pain pathways. This pathway represents an additional analgesic mechanism that can complement its opioid-like effects [17].

The absence of significant Delta-Opioid Receptor (DOR) activity in both mitragynine and corynoxeine indicates that their analgesic and neuroprotective effects are mediated through other pathways [12]. DORs are involved in modulating pain, mood, and neuroprotection, but DOR agonists are associated with the risk of convulsions and other adverse effects.

### 2.2. Pharmacokinetics

Mitragynine, the principal indole alkaloid in Mitragyna speciosa (kratom), exhibits distinctive pharmacokinetic properties that are crucial to its therapeutic applications. It is a lipophilic weak base (pKa ~8.1) with high plasma protein binding (85–95%) and undergoes extensive hepatic metabolism through both phase I (oxidation, demethylation) and phase II (glucuronidation, sulfation) pathways.

The pharmacokinetics of mitragynine demonstrate dose-dependent variations, with key parameters providing insights into its behavior. After a single dose, the Cmax ranged from 17.1 ng/mL (6.65 mg dose) to 125 ng/mL (53.2 mg dose), while at steady-state (Cmax,ss), it ranged from 21.4 ng/mL to 143 ng/mL. The Tmax remained consistent across both single and multiple doses, ranging from 1.0 to 1.7 h. The elimination half-life (t1/2) increased significantly with dose, ranging from 8.5 h at the lowest dose to 43.4 h at the highest dose for single dosing, and from 25.7 to 67.9 h at steady state. Systemic exposure, measured as AUC0–∞, increased proportionally with dose, ranging from 52.8 h·ng/mL to 908 h·ng/mL after a single dose, while steady-state AUC (AUC0–τ,ss) ranged from 85.1 h·ng/mL to 958 h·ng/mL. Clearance (CL) decreased with increasing doses, ranging from 278 L/h at lower doses to 94 L/h at higher doses, indicating nonlinear pharmacokinetics. The volume of distribution (Vd) also increased with dose, from 1349 L to 3788 L for single doses and from 2980 L to 6020 L at steady-state, reflecting extensive tissue distribution [18]. These findings highlight the importance of dose adjustments to account for significant changes in clearance and half-life at higher doses.

A significant factor influencing the pharmacokinetics of mitragynine is the variability between administering pure mitragynine versus raw kratom products. Raw kratom contains a complex mixture of alkaloids and bioactive compounds that can alter absorption, metabolism, and clearance, resulting in greater variability in pharmacokinetic parameters. For instance, other alkaloids in raw kratom may competitively inhibit or enhance mitragynine metabolism, affecting Cmax, Tmax, and t1/2. Furthermore, no standardization of kratom formulations or dosing exists, making it challenging to establish predictable therapeutic outcomes or mitigate potential adverse effects.

The pharmacokinetics of corynoxeine exhibit notable differences between normal physiological conditions and CUMS-induced depression models, which are essential to consider for therapeutic applications. In normal rats, the maximum plasma concentration (Cmax) of corynoxeine was 407.48 ± 10.87 ng/mL, with a time to reach Cmax (Tmax) of 1.67 ± 0.24 h. In contrast, CUMS-induced depression rats demonstrated a lower Cmax of 306.83 ± 18.72 ng/mL and a delayed Tmax of 2.33 ± 0.47 h, indicating reduced absorption under pathological conditions. The elimination half-life (t1/2) was slightly longer in CUMS-induced depression rats (2.68 ± 0.30 h) compared to normal rats (2.40 ± 0.12 h), while the clearance rate (CL) was faster in the depression model (18.06 ± 1.36 L/h/kg versus 14.48 ± 0.61 L/h/kg in normal rats), suggesting a reduced systemic exposure to the compound [14].

Additionally, the volume of distribution (Vd) was significantly higher in CUMS-induced depression rats (69.15 ± 3.25 L/kg) compared to normal rats (50.09 ± 2.11 L/kg), implying greater tissue penetration in the pathological state. The area under the concentration–time curve (AUC), representing overall drug exposure, was reduced in CUMS-induced rats, with AUC0-t values of 1202.97 ± 39.79 ng·h/mL versus 1495.62 ± 55.23 ng·h/mL in normal rats and AUC0-∞ values of 1614.48 ± 119.62 ng·h/mL versus 1914.65 ± 95.66 ng·h/mL, respectively [14]. These findings highlight a significant impact of depression on corynoxeine’s pharmacokinetics, including reduced absorption, faster clearance, and greater distribution into tissues.

The altered pharmacokinetic parameters in the depression model underscore the influence of pathological conditions on the behavior of corynoxeine within the body. These differences emphasize the need for tailored dosing regimens in clinical applications to ensure therapeutic efficacy and safety in patients with underlying conditions. Understanding these variations provides a foundation for optimizing the use of these compounds in different physiological and pathological contexts. Furthermore, these disparities underscore the necessity of further research to optimize standardization, characterize dose–response relationships, and establish regulatory frameworks for the safe and effective clinical use of mitragynine and kratom products.

## 3. Discussion

### 3.1. Summary of Findings

Mitragynine and corynoxeine exhibit significant potential in effectively managing and treating various types of pain through their multifaceted mechanisms of action. When addressing pain management, there is a large basis of pharmacotherapy because of the vastly diverse pathophysiology. While these compounds hold significant potential in neuropathic pain (chemotherapy-induced peripheral neuropathy (CIPN), diabetic neuropathy, postherpetic neuralgia), inflammatory pain, nociceptive pain (post-surgical, musculoskeletal), visceral pain (IBS, endometriosis), and central pain syndromes (MS, spinal cord injury), cancer pain is the most notable for effective management.

Cancer pain presents a unique challenge in pain management due to its complex and multifaceted nature, often involving a combination of nociceptive, neuropathic, and inflammatory components. Traditional opioids like morphine have long been the mainstay for cancer pain management; however, their use is associated with significant drawbacks, including the risk of tolerance, dependence, respiratory depression, and constipation. Furthermore, recent studies have suggested that morphine and other opioids may promote tumor growth and metastasis by enhancing angiogenesis and suppressing immune function [19,20]. This potentially adverse effect on cancer progression highlights the need for alternative analgesics that do not compromise cancer treatment outcomes.

Mitragynine and corynoxeine offer promising alternatives to traditional opioids for cancer pain management. Mitragynine, a partial agonist at mu-opioid receptors, provides effective analgesia while reducing the risk of severe side effects associated with full opioid agonists [16]. Its lack of beta-arrestin 2 recruitment minimizes the risks of respiratory depression, constipation, and tolerance development. Additionally, mitragynine’s partial agonist activity at 5-HT1A receptors and antagonist activity at 5-HT2A receptors may offer mood-enhancing and anxiolytic benefits, which are crucial for the overall well-being of cancer patients [10,21].

Corynoxeine, with its potent anti-inflammatory and neuroprotective properties, can further enhance pain relief in cancer patients. Its ability to reduce inflammation and protect neurons from damage is particularly beneficial in managing pain caused by tumor growth and metastasis [8,22]. Corynoxeine’s vasorelaxant effects, mediated through calcium channel blocking and adrenergic receptor antagonism, can help alleviate pain associated with tumor-induced vascular tension and improve blood flow to affected tissues [6,22].

Moreover, neither mitragynine nor corynoxeine has been shown to promote tumor growth, making them safer options for cancer patients compared to traditional opioids [23,24]. By offering effective analgesia without the risk of enhancing tumor progression, these compounds represent a significant advancement in the management of cancer pain. The combined pharmacological actions of mitragynine and corynoxeine, including their opioid receptor modulation, anti-inflammatory effects, and neuroprotective properties, position them as superior alternatives to traditional opioids like morphine for cancer pain management. Their potential to provide comprehensive pain relief while minimizing adverse effects and supporting overall patient health underscores their importance in the evolving landscape of cancer pain therapeutics.

### 3.2. β-Arrestin Activity

G-protein-coupled receptors such as the opioid receptors are transmembrane receptors that are capable of recruiting proteins β-arrestins to initiate separate cellular signal transduction pathways [12]. One of the most promising aspects of the major alkaloids found in *Mitragyna speciosa* is their lack of beta-arrestin 2 recruitment. Traditional opioids like morphine and fentanyl not only activate G-protein signaling pathways to provide pain relief but also recruit beta-arrestin 2. This recruitment is associated with many adverse effects, such as respiratory depression, constipation, and the development of tolerance and dependence. Beta-arrestin 2 mediates these effects by desensitizing the receptors, internalizing them, and initiating alternative signaling cascades that contribute to these unwanted outcomes [25]. Mitragynine, however, acts as a partial agonist at the mu-opioid receptor without recruiting beta-arrestin 2. This G-protein biased agonism allows it to deliver effective analgesia while minimizing the risks of severe side effects.

By avoiding beta-arrestin 2 pathways, mitragynine reduces the likelihood of respiratory depression, which is the primary cause of fatal opioid overdoses. Additionally, the absence of beta-arrestin 2 recruitment can lead to less constipation and reduced potential for developing tolerance and dependence, making it a safer long-term option for pain management [26].

The dual mechanism of mu-opioid receptor agonism and kappa-opioid receptor antagonism, coupled with the lack of beta-arrestin 2 recruitment, may help to separate the beneficial analgesic effects from the harmful side effects typically associated with opioid therapy. This signaling bias is crucial for developing safer analgesic drugs with fewer side effects. By preferentially activating G-protein pathways and avoiding beta-arrestin 2, these compounds offer a novel approach to pain management that could mitigate the opioid crisis. This ability to separate analgesic efficacy from adverse side effects represents a significant advancement in the pharmacotherapy of pain, especially for chronic conditions that require long-term treatment. Additionally, this selective receptor profile enhances the safety of mitragynine for long-term use in pain management, making it a more reliable option for patients who need consistent and effective pain relief without the risk of convulsions.

### 3.3. Mechanisms Underlying Stimulant Properties

Many stimulants enhance the release or activity of neurotransmitters like dopamine and norepinephrine, which elevate mood. Improved mood can change how patients experience and cope with pain, as negative emotions often heighten pain perception. Pain is both a physical and emotional experience, so by reducing emotional distress, stimulants can lessen perceived pain intensity.

Mitragynine has been reported to have significant stimulant properties, particularly at lower doses. From the reviewed literature, it can be deduced that these stimulant properties are due to its interactions with various neurotransmitter systems [19,27]. As a partial agonist at 5-HT1A receptors, mitragynine may enhance mood and reduce anxiety, contributing to increased energy and alertness [10]. Its affinity for dopamine D2 receptors suggests it could modulate the dopaminergic system, promoting feelings of euphoria and motivation [21]. Additionally, mitragynine’s action as an agonist at alpha-2 adrenergic receptors could lead to sympathomimetic effects such as increased release of norepinephrine, which enhances wakefulness and physical energy [13]. These combined actions indicate potential for improved cognitive function, enhanced mood, and increased physical activity, positioning mitragynine as a unique compound with both stimulant and analgesic properties.

### 3.4. Clinical Considerations

The pharmacokinetic (PK) data for corynoxeine derived from preclinical studies in rats provides critical insights for determining appropriate clinical dosing regimens in humans. To extrapolate these findings to humans, allometric scaling was applied. Using a standard Km factor (6 for rats and 37 for humans), the rat dose of 25 mg/kg converts to approximately 4.05 mg/kg in humans. For an average 70 kg human, this translates to a total dose of approximately 283 mg. However, human dosing must account for differences in absorption and bioavailability, which are often lower in humans compared to animal models. Table 3 presents the human equivalent dose (HED) calculations based on body surface area (BSA) for various animal species, derived from standard allometric scaling methods (Table 3).

For clinical use, an initial regimen could involve split dosing to maintain steady-state plasma levels, given the relatively short t1/2 of ~2.5 h in rats. Administering a 283 mg daily dose as 100 mg every 8 h would ensure consistent plasma concentrations while minimizing peaks and troughs that could affect efficacy or safety. Furthermore, close monitoring of plasma concentrations and therapeutic responses will be critical to optimize dosing, ensure efficacy, and avoid potential toxicity in human trials. These data significantly reduce the guesswork in determining clinical dosages for achieving therapeutic outcomes, such as tumor volume reduction and pain control, assuming similar PK profiles for mitragynine and corynoxeine [14].

Tissue distribution studies demonstrate that corynoxeine predominantly accumulates in the intestine and stomach, with peak concentrations in the intestine reaching 1737.96 ± 128.32 ng/mL in normal rats and 1866.59 ± 163.72 ng/mL in CUMS-induced depression rats at 1 h post-administration. In contrast, significantly lower concentrations were observed in the kidney, lung, and liver, while brain concentrations were negligible. This suggests that corynoxeine has limited permeability across the blood–brain barrier (BBB), likely due to its physicochemical properties, including high lipophilicity.

The limited BBB permeability is an important pharmacological attribute that minimizes corynoxeine’s central nervous system (CNS) activity, reducing the risk of psychotropic effects commonly associated with abuse potential. Despite its high lipophilicity, which aids in distribution across peripheral tissues, its inability to effectively penetrate the CNS suggests a low likelihood of euphoria or reinforcement behaviors typically driving substance misuse. This characteristic makes corynoxeine a safer candidate for therapeutic applications in managing pain and inflammation without the heightened risk of dependency observed with drugs that readily cross the BBB.

### 3.5. Limitations

While mitragynine and corynoxeine exhibit promising potential as safer alternatives to traditional opioids, it is crucial to address their limitations and negative aspects to provide a balanced perspective.

#### 3.5.1. Potential for Abuse and Dependence

Mitragynine, particularly at higher doses, can exhibit psychotropic effects, which may lead to misuse and recreational abuse [28]. While it may have a lower abuse potential compared to traditional opioids, the risk remains. There are emerging reports and anecdotal evidence of individuals developing kratom use disorder (KUD). This dependence can lead to withdrawal symptoms upon cessation, similar to traditional opioids and opioid use disorder (OUD) [29,30].

Mitigating the potential for abuse and dependence on kratom-derived alkaloids like mitragynine and corynoxeine requires a multifaceted approach, especially given their psychotropic effects and potential for KUD. Implementing regulatory measures to standardize product quality and ensure purity across kratom products would be essential to reduce variability in potency, which can contribute to misuse. Clear labeling on dosage and potential risks, along with public education on safe usage, could help guide responsible use. Additionally, further research into formulating kratom products with lower abuse potential, such as compounds that lack full opioid receptor activation, could support safer therapeutic applications. Encouraging supervised use under medical guidance for patients with chronic pain could also help manage dependency risks while maintaining therapeutic efficacy [31,32].

#### 3.5.2. Lack of Standardization

The potency of mitragynine and corynoxeine can vary significantly depending on the source and preparation of kratom. This variability can lead to inconsistent therapeutic outcomes and difficulties in dosing [33,34]. Another significant challenge in the clinical application of mitragynine and corynoxeine is the presence of diastereomers and stereoisomers, which complicates their purification and isolation. Both mitragynine and corynoxeine exist in multiple stereoisomeric forms, each with potentially different pharmacological profiles and potencies [2,6]. The stereochemistry of these compounds affects their interaction with biological targets, leading to variability in therapeutic efficacy and side effects. For instance, mitragynine has diastereomers which differ in their spatial configuration. These structural variations can result in significant differences in receptor binding affinity and activity, impacting the overall effectiveness of the kratom extracts [32].

The extraction, isolation, and purification processes for these compounds must account for the separation of these isomers, which often requires sophisticated and costly techniques such as chiral chromatography. Additionally, the presence of multiple isomers in raw extracts can lead to batch-to-batch variability, making standardization difficult [35,36]. This variability poses challenges not only for clinical dosing but also for ensuring consistent quality in commercially available products. Without rigorous quality control measures, the therapeutic potential of mitragynine and corynoxeine could be compromised by the presence of less active or inactive isomers, reducing the overall efficacy and safety of the treatment. Unfortunately, many studies present in the literature do not take the extra steps to ensure the separation of these isomers. Instead, they often cover only the basic characteristics that are consistent across similar compounds, leading to an incomplete understanding of the individual isomers’ effects [35,36]. The complexity introduced by these stereoisomers necessitates advanced analytical methods and comprehensive regulatory standards to ensure that only high-quality, standardized preparations are used in clinical settings.

#### 3.5.3. Adverse Drug Effects

Long-term safety studies of kratom alkaloids, particularly mitragynine, suggest a relatively favorable profile compared to classical opioids. Human studies indicate no significant hematological, biochemical, or organ toxicity at typical doses of kratom use. Unlike opioids, mitragynine does not cause substantial respiratory depression or severe withdrawal symptoms. However, at higher doses or prolonged use, there is some evidence of cardiovascular risks, such as potential QT interval prolongation, which warrants further investigation. Additionally, while physical dependence is less pronounced than with opioids, mild withdrawal symptoms have been reported, including irritability, insomnia, and muscle aches, which are typically self-limiting. These findings underscore the need for additional long-term clinical studies to better understand the chronic safety profile and potential risks associated with prolonged use of kratom and its alkaloids [37].

Although mitragynine may cause less constipation compared to full opioid agonists, some users report gastrointestinal disturbances such as nausea and vomiting. These side effects can impact patient comfort and adherence to treatment regimens [38].

There are limited data on the cardiovascular effects of long-term use of mitragynine and corynoxeine. Some studies suggest potential alterations in blood pressure and heart rate, which warrant further investigation [39]. These potential changes could pose risks, particularly for patients with pre-existing cardiovascular conditions. Mitragynine at 10 μM inhibits the hERG current, prolongs action potential duration (APD), and induces arrhythmia. It also suppresses inward potassium currents (IKr) in hiPSC-CMs, contributing to its cardiotoxic effects [39,40]. This cardiotoxicity requires further investigation to fully understand the risk profile.

Some case reports and studies have indicated that mitragynine and other kratom alkaloids can cause elevations in liver enzymes, indicating potential liver damage [41]. This hepatotoxicity is a concern, particularly for long-term users or those with pre-existing liver conditions. In rare cases, severe liver damage leading to acute liver failure has been reported, necessitating close monitoring of liver function in patients using these compounds [42].

One notable pharmacological interaction of corynoxeine is its ability to induce the expression of cytochrome P450 3A4 (CYP3A4) enzymes. This induction can significantly alter the metabolism of co-administered drugs that are CYP3A4 substrates, potentially reducing the efficacy of these drugs due to faster metabolism or, in some cases, leading to subtherapeutic levels [42,43]. In contrast, mitragynine has been shown to inhibit multiple CYP enzymes, including CYP2D6/9 and CYP3A4/5, which may elevate the plasma concentrations of drugs metabolized by these pathways, increasing the risk of adverse effects or toxicity [39,44].

This differential modulation of CYP3A4 between corynoxeine (induction) and mitragynine (inhibition) presents a unique challenge, especially for patients taking multiple medications with narrow therapeutic indices. For instance, the inhibition of CYP3A4 by mitragynine could result in elevated levels of drugs like anticoagulants, certain immunosuppressants, or anticonvulsants, which depend on CYP3A4 for clearance, thus increasing the likelihood of toxicity. Conversely, corynoxeine’s induction of CYP3A4 may reduce the plasma levels of these medications, risking therapeutic failure.

Given the prevalence of polypharmacy among pain management patients, these interactions underscore the need for careful monitoring when kratom-derived compounds are co-administered with other medications. Although data are limited, the potential for pharmacodynamic interactions—such as enhanced sedation with opioids or benzodiazepines, or increased risk of serotonin syndrome with SSRIs—further highlights the complexity of drug–drug interactions with kratom alkaloids. These interactions call for additional research and clinical awareness to manage the potential risks associated with concurrent kratom use. A summary of adverse drug reactions, including gastrointestinal effects, cardiovascular effects, hepatotoxicity, and drug interaction challenges (e.g., toxicity and efficacy), can be found in Figure 3.

In summary, mitragynine- and corynoxeine-related alkaloids hold potential as safer alternatives to traditional opioids but are limited by challenges such as the potential for abuse, dependence, and inconsistent potency due to stereoisomeric variability. Mitragynine may lead to kratom use disorder (KUD), with psychotropic effects and withdrawal symptoms, while both compounds exhibit adverse effects including gastrointestinal disturbances, cardiovascular risks, and hepatotoxicity. Their differential modulation of CYP3A4—mitragynine as an inhibitor and corynoxeine as an inducer—complicates polypharmacy, increasing risks of toxicity or therapeutic failure. Rigorous research, advanced standardization techniques, and regulatory oversight are necessary to address these limitations and ensure their safe and effective therapeutic use.

## 4. Methods

### 4.1. Data Sources

The following scientific databases were utilized as tertiary resources in reviewing the relevant literature: PubMed Central^®^, ScienceDirect, Scopus, BIOSIS Previews, Embase, OpenAIRE, PsycINFO, CINAHL, LILACS, Google Scholar and ThaiLIS. The search criteria included clinical and scientific literature from all languages published between June 1950 and July 2024. Combinations of the following Medical Subject Headings (MeSH) were employed: *Mitragyna speciosa*, kratom, mitragynine, corynoxeine, beta-arrestin, opioid, dose, toxicity, and agonist.

Additionally, the search results were supplemented by references obtained from recent reviews and citations from the search returns.

### 4.2. Inclusion and Exclusion Criteria

#### 4.2.1. Inclusion Criteria:

Study Focus: All studies reporting outcomes related to the pharmacology, pharmacokinetics, and therapeutic effects of indole and oxindole alkaloids, specifically Mitragyna speciosa (kratom) and its constituents mitragynine and corynoxeine.

Study Types: Human, animal, and in vitro studies.

Formulation and Administration: Any formulation (e.g., crude extracts, isolated alkaloids) and administration route (e.g., oral, intravenous).

Outcome Measures: Studies must report specific clinical, pharmacological, or toxicological outcomes, including but not limited to efficacy, safety, metabolic pathways, receptor binding, and signaling mechanisms.

#### 4.2.2. Exclusion Criteria:

Type of Publication: Studies not reporting on specific clinical or experimental outcomes, such as commentaries, letters to editors, review articles, and abstracts, without full data.

Irrelevant Focus: Studies not focusing on the alkaloids of interest or not addressing their pharmacological or therapeutic implications.

Insufficient Data: Studies lacking adequate methodological details or outcome data necessary for analysis.

### 4.3. Study Selection

The original search yielded 492 reports. A.A., M.H., C.D., and H.M. independently examined each by title and abstract. After eliminating studies with exclusionary criteria, 51 papers met our original search criteria which included basic science, preclinical studies, case reports, observational studies, systematic reviews and meta-analyses.

### 4.4. Data Extraction

A standardized data extraction process was used that stratified study identification, publication details, study design, population characteristics, interventions, comparators, outcomes, and methodological quality indicators. Detailed information about the study population was extracted, including species (for animal studies), cell lines (for in vitro studies), demographic characteristics (for human studies), sample size, and relevant baseline characteristics. Key outcomes of interest were documented, including pharmacological effects, pharmacokinetics, therapeutic efficacy, adverse effects, receptor binding, and signaling mechanisms. Both primary and secondary outcomes were included to provide a comprehensive overview. Quality indicators were assessed for each study, including study design (randomized controlled trials, observational studies, in vitro experiments), risk of bias, and reporting standards.

Tools such as the Cochrane Risk of Bias tool and the Newcastle–Ottawa Scale were employed where appropriate. Extracted data were entered into a database for synthesis. Discrepancies between reviewers were resolved through discussion, and if necessary, a third reviewer was consulted to achieve consensus.

This rigorous approach to data extraction ensured that all relevant information was captured systematically and consistently, providing a robust foundation for the subsequent analysis and synthesis of the findings in this review.

## 5. Conclusions

This systematic review explores the potential of mitragynine and corynoxeine, selected as two key alkaloids from the diverse indole and oxindole classes, as possible alternatives to traditional opioids for pain management. These alkaloids from *Mitragyna speciosa* (kratom) showcase distinct pharmacological profiles and structural features, illustrating the broader pharmacodynamic diversity present within indole and oxindole compounds. Their mechanisms of action could be relevant in addressing various types of pain, including neuropathic, inflammatory, nociceptive, visceral, and central pain syndromes. The pharmacological profile of mitragynine, particularly its partial agonism at mu-opioid receptors and interactions with serotonin and dopamine receptors, suggests an analgesic effect with potentially fewer side effects—such as reduced risk of respiratory depression and dependence—compared to conventional opioids. Similarly, corynoxeine’s reported anti-inflammatory, neuroprotective, and vasorelaxant effects may contribute to pain relief in specific contexts, although its variable presence in kratom products and relatively low concentration limit its consistent impact.

Several challenges must be addressed to better understand the therapeutic viability of these compounds. Variability in the potency and quality of kratom products due to the presence of stereoisomers and lack of standardization complicates clinical application, and the potential for adverse effects such as gastrointestinal disturbances, cardiotoxicity, and hepatotoxicity requires further investigation. Additionally, the interaction of indole and oxindole alkaloids with cytochrome P450 enzymes underscores the need for understanding drug–drug interactions to ensure safe co-administration with other medications.

Future research should prioritize comprehensive clinical trials to assess long-term safety and efficacy, detailed mechanistic studies to elucidate their pharmacodynamic and pharmacokinetic profiles, and the development of standardized quality control measures to ensure consistent and reliable preparations. Expanding research to include other alkaloids within the indole and oxindole classes may also provide valuable insights into the therapeutic scope of these diverse compounds. Collaborative efforts among researchers, regulatory agencies, and public health policymakers are essential to navigate regulatory and ethical challenges, enabling a rigorous and safe approach to exploring these compounds’ role in clinical settings. Previous analyses have also emphasized that scheduling kratom under the Controlled Substances Act (CSA) would criminalize its use and possession and create significant barriers to research, and could lead to serious public health repercussions, including potentially thousands of drug overdose fatalities. As such, including kratom in the CSA is not advised.

In summary, while indole and oxindole alkaloids demonstrate potential as components in pain management strategies, further rigorous research is needed to substantiate their clinical utility and ensure safe application. Their unique pharmacological characteristics and varied mechanisms of action highlight their promise but also reinforce the necessity for cautious and well-regulated investigation before they may be integrated into therapeutic practice.

## Figures and Tables

**Figure 1 pharmaceuticals-18-00222-f001:**
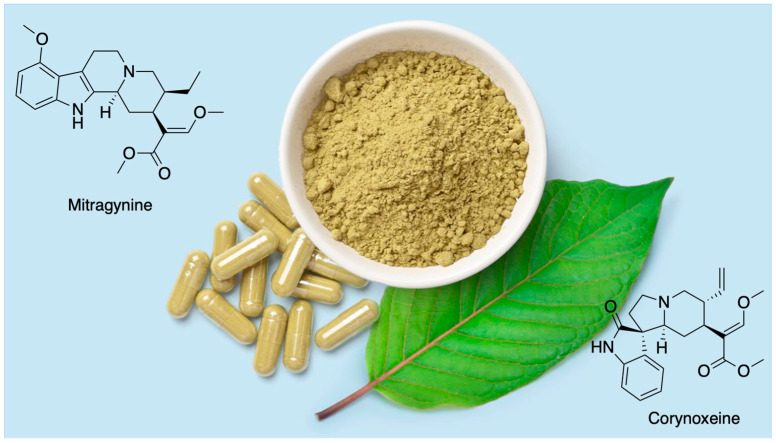
*Mitragyna speciosa*, commonly known as kratom, contains bioactive secondary metabolites, specifically indoles (e.g., mitragynine) and oxindoles (e.g., corynoxeine).

**Figure 2 pharmaceuticals-18-00222-f002:**
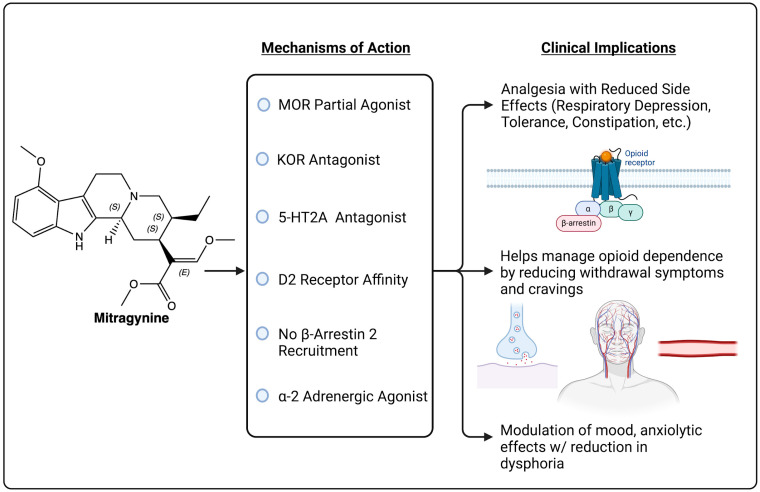
Polypharmacology of mitragynine, the principal indole alkaloid derived from *Mitragyna speciosa*.

**Figure 3 pharmaceuticals-18-00222-f003:**
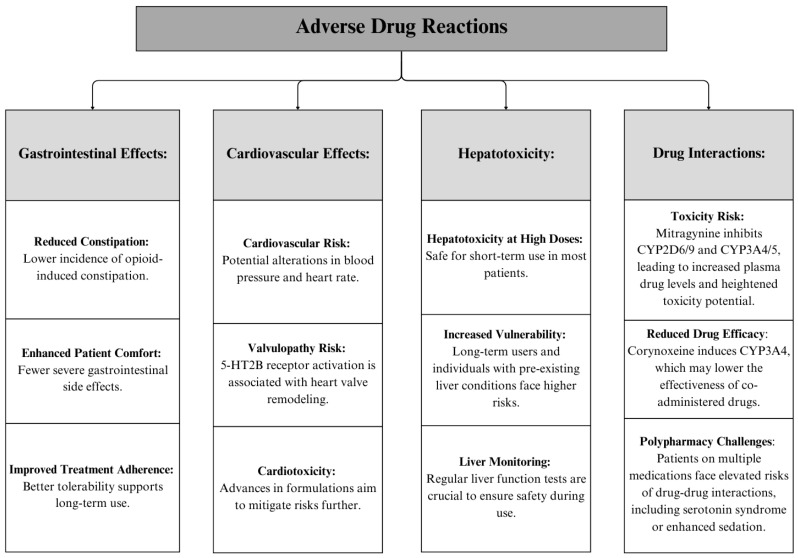
Summary of adverse drug reactions associated with mitragynine and corynoxeine.

**Table 1 pharmaceuticals-18-00222-t001:** Principal structures of oxindole alkaloids of *Mitragyna speciosa*.

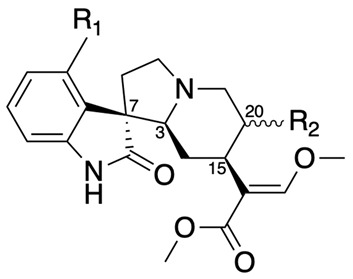						
Compound	R_1_	3	7	15	20	R_2_
Corynoxeine	H	S	R	S	R	CHCH_2_
Corynoxine A	H	S	S	S	S	CH_2_CH_3_
Corynoxine B	H	S	R	S	S	CH_2_CH_3_
Mitrafoline	OH	S	S	S	S	CH_2_CH_3_
Speciofoline	OH	R	R	S	S	CH_2_CH_3_
Specionoxeine	OCH_3_	S	R	S	R	CHCH_2_
Rhynchophylline	H	S	R	S	R	CH_2_CH_3_
Rotundifoline	OH	S	S	S	R	CH_2_CH_3_
Rotundifoleine	OH	S	S	S	R	CHCH_2_

**Table 2 pharmaceuticals-18-00222-t002:** Principal structures of indole alkaloids of *Mitragyna speciosa*.

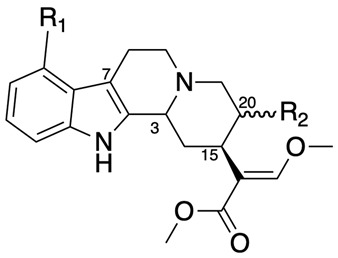					
Compound	R_1_	3	15	20	R_2_
Mitragynine	OCH_3_	S	S	S	CH_2_CH_3_
Speciogynine	OCH_3_	S	S	R	CH_2_CH_3_
Speciociliatine	OCH_3_	R	S	S	CH_2_CH_3_
Mitraciliatine	OCH_3_	R	S	R	CH_2_CH_3_
Paynantheine	OCH_3_	S	S	R	CHCH_2_
Corynantheidine	OH	S	S	S	CH_2_CH_3_

**Table 3 pharmaceuticals-18-00222-t003:** Human equivalent dose calculation based on body surface area.

Species	Reference Body Weight	Body Surface Area (m^2^)	To Convert Dose in mg/kg to Dose in mg/m^2^, Multiply by Km	To Convert Animal Dose in mg/kg to HED in mg/kg, Either	
				Divide Animal Dose by	Multiply Animal Dose by
Human	60	1.62	37	-	-
Mouse	0.02	0.007	3	12.3	0.081
Hamster	0.08	0.016	5	7.4	0.135
Rat	0.15	0.025	6	6.2	0.162
Ferret	0.3	0.043	7	5.3	0.189
Guionea pig	0.4	0.05	8	4.6	0.216
Rabbit	1.8	0.15	12	3.1	0.324
Dog	10	0.5	20	1.8	0.541
Monkeys (rhesus)	3	0.25	12	3.1	0.324
Marmoset	0.35	0.06	6	6.2	0.162
Squirrel Monkey	0.6	0.09	7	5.3	0.189
Baboon	12	0.6	20	1.8	0.541
Micro Pig	20	0.74	27	1.4	0.730
Mini Pig	40	1.14	35	1.1	0.946

Data obtained from FDA draft guidelines. FDA: Food and Drug Administration, HED: Human equivalent dose.

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
