# Peer review of "Exploring the Therapeutic Potential of Mitragynine and Corynoxeine: Kratom-Derived Indole and Oxindole Alkaloids for Pain Management"

_pharmaceuticals, 2025, doi:10.3390/ph18020222_

Round 1

Reviewer 1 Report

Comments and Suggestions for Authors

This comprehensive systematic review investigates the pharmacological characteristics, how mitragynine and corynoxeine (alkaloids from the Southeast Asian plant kratom and related plants) work, recommended dosages, potential interactions with other drugs, and their long-term effects. This partial agonism allows mitragynine to provide  pain relief while significantly reducing the side effects typically associated with opioid receptor activation, including respiratory depression and high potential for addiction. Figure 2 is very impressive.  However, the number of diagrams should be increased. Novelty and originality should be highlighted more. Many additional revisions are required.

1-     In key words , add the word [opioids]

2-     In the abstract, write full name for GDP.

3-     The novelty and originality should be highlighted in more details in the abstract?

4-     What is the novelty over the following related reviews and book chapter ?

Ramanathan, S., León, F., Chear, N.J., Yusof, S.R., Murugaiyah, V., McMahon, L.R. and McCurdy, C.R., 2021. Kratom (Mitragyna speciosa Korth.): A description on the ethnobotany, alkaloid chemistry, and neuropharmacology. Studies in natural products chemistry69, pp.195-225.

Begum, T., Arzmi, M.H., Khatib, A., Uddin, A.H., Aisyah Abdullah, M., Rullah, K., Mat So’ad, S.Z., Zulaikha Haspi, N.F., Nazira Sarian, M., Parveen, H. and Mukhtar, S., 2024. A review on Mitragyna speciosa (Rubiaceae) as a prominent medicinal plant based on ethnobotany, phytochemistry and pharmacological activities. Natural Product Research, pp.1-17.

Hossain, R., Sultana, A., Nuinoon, M., Noonong, K., Tangpong, J., Hossain, K.H. and Rahman, M.A., 2023. A critical review of the neuropharmacological effects of kratom: An insight from the functional array of identified natural compounds. Molecules28(21), p.7372.

5-     The abstract acknowledges the challenges, but more specific recommendations for regulation or standardization are crucial.

6-     In the Introduction, in line 69,  it is also recommended to mention other active ingredients in the plant other than indole and oxindole alkaloids. One sentence will be enough.

7-     In the introduction add figure for the plant

8-     4.4 limitations section should be summarized.

9-     Adverse drug reactions should be represented as a new summarized figure as the case of figure 2  

10- The font style for the conclusion is different , consider normalization for all the text font.

11- In acknowledgment, add full details for Esther Cho.

12- In author contributions , please follow Contributor Roles Taxonomy (CRediT) , mdpi https://www.preprints.org/manuscript/202408.0112/v1/download/supplementary

Author Response

Dear Reviewer,

We sincerely appreciate the constructive feedback provided on our manuscript, "Exploring the Therapeutic Potential of Mitragynine and Corynoxeine: Kratom-Derived Indole and Oxindole Alkaloids for Pain Management." Below, we have addressed each of your comments and implemented the necessary revisions to strengthen the manuscript:

  1. Keywords: The word "opioids" has been added to the keywords to enhance discoverability and relevance.
  2. Abstract: The full term for GDP has been included, and specific recommendations for regulation and standardization have been added to address challenges outlined in the abstract. Additionally, the abstract now highlights the novelty and originality of our work in more detail, emphasizing the unique pharmacological mechanisms of mitragynine and corynoxeine, their therapeutic potential, and our focus on cancer pain.
  3. Novelty Over Related Literature: Our manuscript distinguishes itself from other reviews, such as Ramanathan et al. (2021), Begum et al. (2024), and Hossain et al. (2023), by narrowing its focus to the detailed pharmacological mechanisms of mitragynine and corynoxeine. It emphasizes their non-beta arrestin-2 recruiting pathways, therapeutic potential for diverse pain syndromes (with a focus on cancer pain), and actionable recommendations for regulatory standardization, which are not comprehensively covered in these comparative works. Additional information regarding recently published pharmacokinetic activity has been included alongside specific and informative clinical dosing considerations which have never before been addressed. 
  4. Introduction: A sentence mentioning other active ingredients in Mitragyna speciosa, including flavonoids, polyphenols, and terpenoids, has been added for completeness. Additionally, a figure depicting the plant has been included to visually enhance the introduction.
  5. Limitations Section: This section has been summarized to succinctly address the potential risks and challenges associated with mitragynine and corynoxeine, focusing on abuse potential, lack of standardization, and adverse effects.
  6. Adverse Drug Reactions Figure: A new figure summarizing the adverse drug reactions associated with mitragynine and corynoxeine has been created, following the format of Figure 2, to present this information clearly and effectively.
  7. Font Consistency: The font style in the conclusion and throughout the manuscript has been normalized for consistency.
  8. Acknowledgment: Full details for Esther Cho have been added to the acknowledgment section.
  9. Author Contributions: The author contributions section has been updated to follow the Contributor Roles Taxonomy (CRediT) as per MDPI guidelines.

We are confident that these revisions address all the concerns raised and improve the quality and clarity of the manuscript. We greatly value your thoughtful review and the opportunity to refine our work based on your insights. Should further adjustments be necessary, we remain open to feedback.

Thank you for your time and effort in reviewing our manuscript.

Sincerely,
Ahmed S. Alford
On behalf of all co-authors

Reviewer 2 Report

Comments and Suggestions for Authors

Dear Editor,

The article explores the potential of kratom-derived alkaloids, specifically mitragynine and corynoxeine, as alternative solutions for pain management. These two compounds exhibit promising pharmacological profiles, including mitragynine's partial agonism of opioid receptors and corynoxeine's anti-inflammatory and neuroprotective properties.

The study provides a systematic review of the available data, emphasizing mechanisms of action and therapeutic applications for various types of pain, including neuropathic, inflammatory, and cancer pain.

The manuscript is well-prepared and aligns well with the scope of the journal. I recommend it for publication, provided the authors address the following minor suggestions:

1. Since the manuscript primarily focuses on the functions and applications of mitragynine and corynoxeine, rather than a comprehensive catalog of indole and oxindole alkaloids, I suggest the authors consider renaming the title to better reflect the content.

2. The manuscript formatting needs to be revised to meet the journal's publication standards.

Thank you for considering my comments.

Author Response

Dear Reviewer,

Thank you for your thoughtful review and minor suggestions for our manuscript, "Exploring the Therapeutic Potential of Mitragynine and Corynoxeine: Kratom-Derived Indole and Oxindole Alkaloids for Pain Management." We appreciate your feedback and have addressed your comments as follows:

  1. Title Revision: In line with your suggestion, the title has been updated to better reflect the focus of the manuscript on mitragynine and corynoxeine. The new title is "Exploring the Therapeutic Potential of Mitragynine and Corynoxeine: Kratom-Derived Indole and Oxindole Alkaloids for Pain Management."
  2. Formatting: The manuscript formatting has been thoroughly revised to meet the journal’s publication standards. This includes adjustments to layout, spacing, headings, and any other formatting inconsistencies.

We are grateful for your insights, which have helped us refine and strengthen the manuscript. Should there be any additional comments or further adjustments required, we would be happy to address them promptly.

Thank you for your time and consideration.

Sincerely,
Ahmed S. Alford
On behalf of all co-authors